# Broadband Millimeter-Wave Power Amplifier Using Modified 2D Distributed Power Combining

**Jihoon Kim** 

Department of Electronics Engineering, Pai Chai University, Daejeon 35345, Korea; j7h7.kim@pcu.ac.kr;
Tel.: +82-42-520-5595

**Abstract:** A broadband millimeter-wave (mmWave) power amplifier (PA) was implemented using a modified 2D distributed power combining technique. The proposed power combining was based on a single-ended dual-fed distributed combining (SEDFDC) design technique using zero-phase shifting (ZPS) transmission lines. To improve the input/output power distribution of each power cell within a wide frequency range, $N/2$-way power dividers/combiners were inserted into the distributed combining structure. Modified ZPS lines also simplified the combining structure and curbed phase variation according to the frequency. These modifications enabled power combining cells to increase without degrading the power bandwidth. The proposed PA was fabricated with a commercial 0.15 μm GaAs pseudo high electron-mobility transistor (pHEMT) monolithic microwave-integrated circuit (MMIC) process. It exhibited 20.3 to 24.2 dBm output power ($P_{out}$), 12.9 to 21.8 dB power gain, and 5.2% to 12.7% power-added efficiency (PAE) between 26 and 56 GHz.

**Keywords:** broadband; millimeter-wave; distributed power combining

## 1. Introduction

The commercialization of 5G communication has extended the carrier frequency to the millimeter wave band and requires a radio frequency (RF) beamforming system consisting of array antennas and transceiver chains. Transceiver chains include millimeter-wave power amplifiers (PAs), low noise amplifiers (LNAs), phase shifters, mixers, etc. Figure 1 shows a block diagram of 1 × 4 array antennas and transmitter chains as an example mainly used in 5G communication systems. As shown in Table 1, the 3rd Generation Partnership Project (3GPP) has already defined equivalent isotropic radiated power (EIRP) specifications for power class 3 for handheld applications such as smartphones. Plus, it has defined higher EIRP specifications for power classes 1 and 2 that serve vehicles and fixed wireless access (FWA) systems [1]. In the future, it is expected that more power will be required in 5G communications as in existing Long Term Evolution (LTE). Therefore, at present, millimeter-wave PAs in 5G mobile user equipment (UE) are mostly implemented as complementary metal-oxide- semiconductor (CMOS) PAs, but gallium arsenide (GaAs) PAs, which can generate higher output power due to the fact of its high breakdown voltage, is attracting attention again.

Meanwhile, 5G communication requires PAs that can operate at various millimeter-wave frequencies such as 28, 39, and 47 GHz [2]. Because broadband millimeter-wave PAs can achieve this requirement in a single chip, they can reduce system complexity and form factor. Millimeter-wave PAs generally require a power combining technique, because single PAs cannot obtain the desired power level; however, conventional power combining circuits, such as parallel combining techniques using Wilkinson combiners, inevitably reduce power bandwidth due to the lossy power matching from very low optimum loads. Although a series power combining technique using stacked field-effect transistors (FETs) somewhat mitigates this problem, it also possesses some drawbacks such as easy oscillation and a limited number of stacked FETs at high frequencies [3]. Recently, single-ended dual-fed

distributed combining (SEDFDC) designs with zero-phase shifting (ZPS) transmission lines have been proposed for millimeter-wave CMOS PAs as shown in Figure 2a [4,5]. As a similar approach, dual-fed distributed amplifiers (DAs) have been reported to enhance radiated gain and power at the leaky-wave antenna [6,7]. A distributed amplifier (DA) can inherently produce ultra-wideband characteristics. Moreover, ZPS transmission lines can optimize the output power of each power cell in the distributed structure [4–7]. However, References [4,5] still showed a $P_{sat}$ less than 18 dBm and a power bandwidth less than 30% in the millimeter-wave band. References [6,7] were implemented at low frequencies below 4 GHz. In relation to this, this work proposed a modified distributed power-combining structure as shown in Figure 2b.

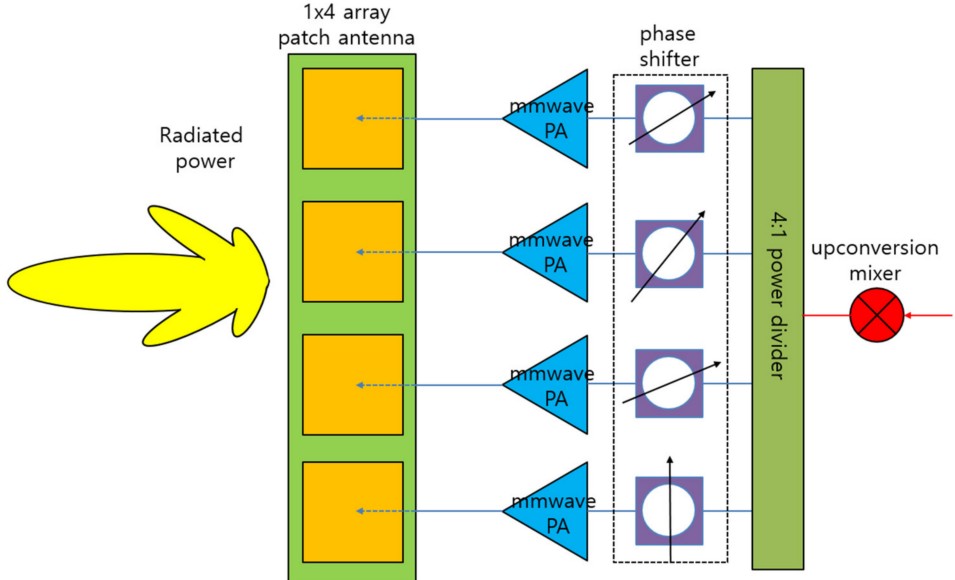

**Figure 1.** Block diagram of $1 \times 4$ array antennas and transmitter chains in 5G communication systems.

**Table 1.** User equipment (UE) minimum peak equivalent isotropic radiated power (EIRP) according to power class and operating band.

| UE Power Class | UE Application | Operating Band | Minimum Peak EIRP [1] (dBm) |
|---|---|---|---|
| 1 | Fixed wireless access (FWA) | n257 (26.5~29.5 GHz) | 40 |
| | | n258 (24.25~27.5 GHz) | 40 |
| | | n260 (37~40 GHz) | 38 |
| | | n261 (27.5~28.35 GHz) | 40 |
| 2 | Vehicle | n257 (26.5~29.5 GHz) | 29 |
| | | n258 (24.25~27.5 GHz) | 29 |
| | | n261 (27.5~28.35 GHz) | 29 |
| 3 | Handheld | n257 (26.5~29.5 GHz) | 22.4 |
| | | n258 (24.25~27.5 GHz) | 22.4 |
| | | n260 (37~40 GHz) | 20.6 |
| | | n261 (27.5~28.35 GHz) | 22.4 |
| 4 | High power non-handheld | n257 (26.5~29.5 GHz) | 34 |
| | | n258 (24.25~27.5 GHz) | 34 |
| | | n260 (37~40 GHz) | 31 |
| | | n261 (27.5~28.35 GHz) | 34 |

[1] EIRP = antenna array factor + single antenna gain + $\log_2$ (number of PA chains) + single PA's $P_{out}$.

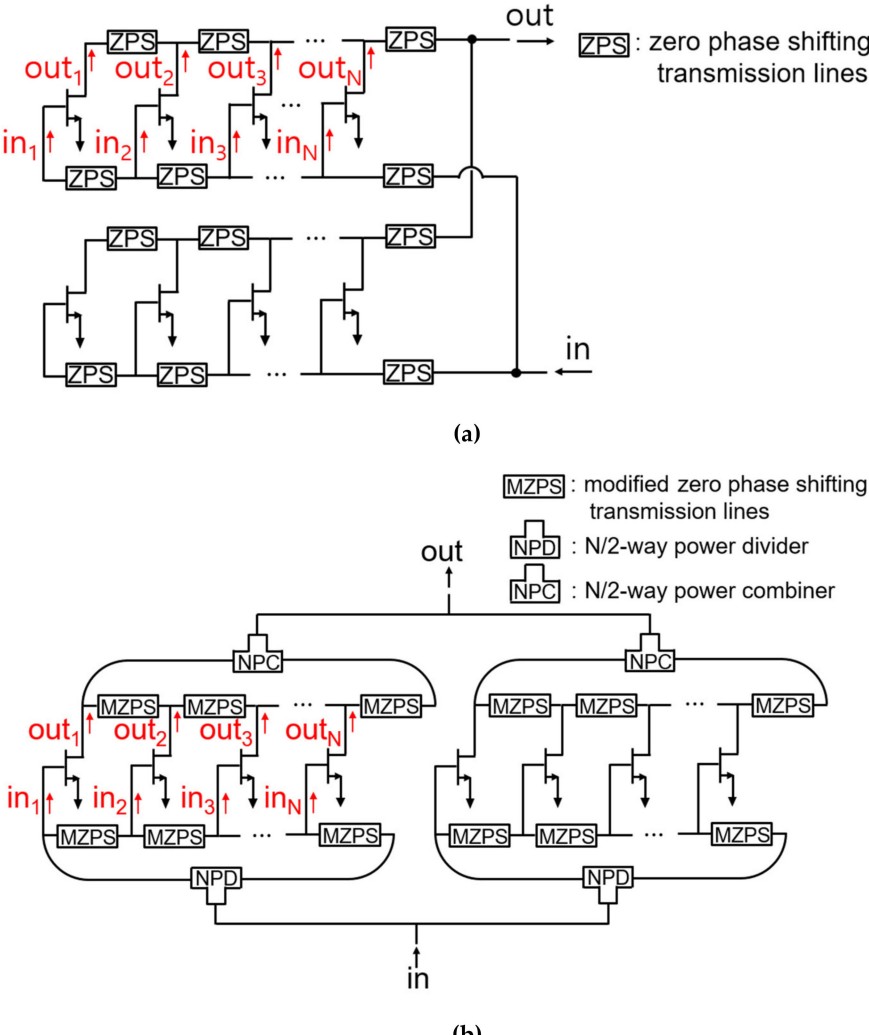

**Figure 2.** Block diagram of $2 \times N$ distributed power-combining structure: (**a**) conventional single-ended dual-fed distributed power-combining and (**b**) proposed distributed power-combining structure.

The proposed structure has the advantage of maintaining the wideband characteristics when more FETs are combined into the type of SEDFDC. The PA monolithic microwave-integrated circuit (MMIC) using the proposed structure exhibits saturated output power ($P_{sat}$) over 20 dBm from the full Ka- and Q-bands to the V-band (26–56 GHz).

## 2. Circuit Design

Figure 3 shows the simulated amplitude and phase differences of the inputs ($in_1$ and $in_N$ in Figure 2a) and the outputs ($out_1$ and $out_N$ in Figure 2a) between the first ZPS power cell and the last, according to the number of cells ($N$) in the SEDFDC structure. In the simulation, 15 dBm of RF power was supplied as the input power. At $N = 2$, each cell had an almost equal input/output signal amplitude and phase through the wideband, while at $N > 2$, these symmetries clearly began to dissolve. This incongruity seriously reduced the power and gain bandwidth. To compensate for this input/output signal mismatch, $N/2$-way power dividers/combiners were inserted into the distributed power-combining structure as shown in Figure 2b. Figure 4 shows the simulated results of the improved input/output amplitude and phase distribution when the 2-way and 3-way power dividers/combiners were each inserted in the case of the $2 \times 4$ combining and $2 \times 6$ combining, respectively. Both the input and output were improved by inserting the $N/2$-way power dividers/combiners.

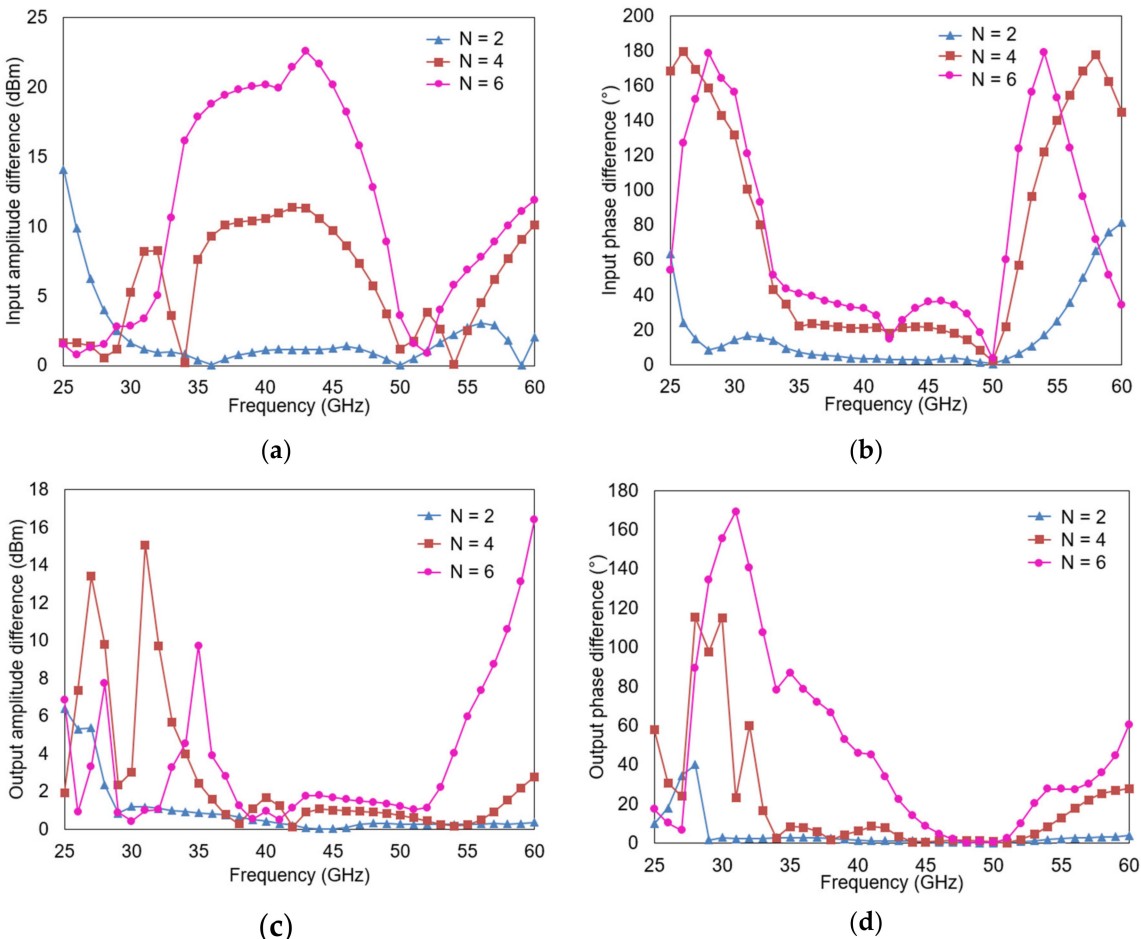

**Figure 3.** Simulated amplitude and phase differences of the inputs and outputs between the first ZPS power cell and the last, according to the number of cells (*N*): (**a**) input amplitude difference (**b**) input phase difference (**c**) output amplitude difference (**d**) output phase difference.

In Figure 3a,b, the input amplitude and phase difference from 25 GHz to 60 GHz were within 3 dB and 80°, respectively, when *N* = 2; but when *N* = 4, they increased to 11 dB and 180°, respectively, and when N = 6, they increased to 22 dB and 180°, respectively. The output amplitude and phase difference were also within 6.5 dB and 40°, respectively, when *N* = 2, as shown in Figure 3c,d, but were 15 dB and 115°, respectively, at *N* = 4 and increased to 16.5 dB and 170°, respectively, at *N* = 6. If *N*/2-way power dividers/combiners were added to the SEDFDC, as shown in Figure 4a–d, the input amplitude and phase difference from 25 GHz to 60 GHz decreased within 8 dB and 120° at *N* = 4, respectively. At *N*, =; 6, they decreased to 5 dB and 30°, respectively. The output amplitude and phase difference also decreased to within 2 dB and 60°, respectively, at *N* = 4 and within 4 dB and 60°, respectively, at *N* = 6. Thus, this improvement increased the output power of SEDFDC in the wideband by uniformly inputting the signal to each power cell and uniformly merging the output power from each power cell.

The ZPS transmission lines are usually designed as a composite right-/left-handed (CRLH) type [8]. The ZPS transmission line can combine the output power of each power cell inside the SEDFDC in phase to maximize the total output power [9]. However, as *N* increases, the layout complexity and DC biasing difficulty increases, and the cross-coupling induced by inductive lines, which consist of ZPS lines, becomes more severe. These factors degrade the power combining efficiency in this structure. Therefore, the ZPS lines need to be modified into a simplified structure without shunt inductances and series capacitances. As shown in Figure 5, the modified ZPS line without a shunt inductance and a series capacitance can be equivalently replaced with a simple microstrip line. Although the

simplified structure deviates from the zero-degree phase, Figure 5 shows that it retards the phase change according to the frequencies.

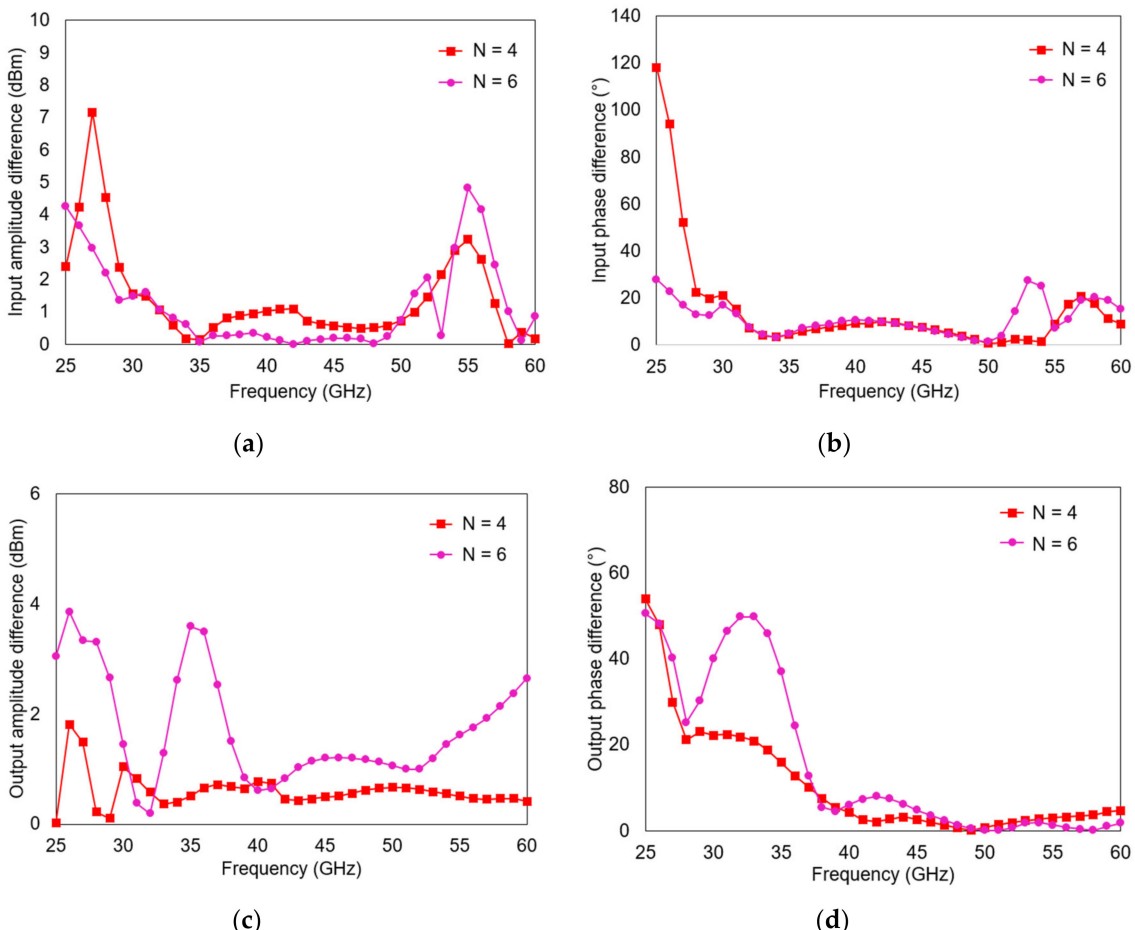

(**a**)

(**b**)

(**c**)

(**d**)

**Figure 4.** Simulated amplitude and phase differences of the inputs and outputs between the first ZPS power cell and the last, according to the number of cells (*N*) when *N*/2-way power dividers/combiners were inserted in the combining structure: (**a**) input amplitude difference (**b**) input phase difference (**c**) output amplitude difference (**d**) output phase difference.

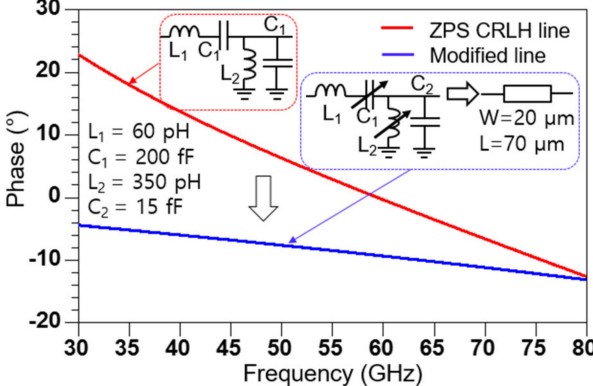

**Figure 5.** Phase characteristics of the ZPS composite right-/left-handed (CRLH) line and modified line.

The modified ZPS lines can extend the overall power bandwidth at the cost of marginally sacrificing the optimum output power at the zero-degree phase frequency. Figure 6 compares the

simulated output power of the proposed structure with that of its conventional counterpart according to the frequencies. The proposed PA exhibited a greater amount of wideband power characteristics than the conventional one without critical power degradation.

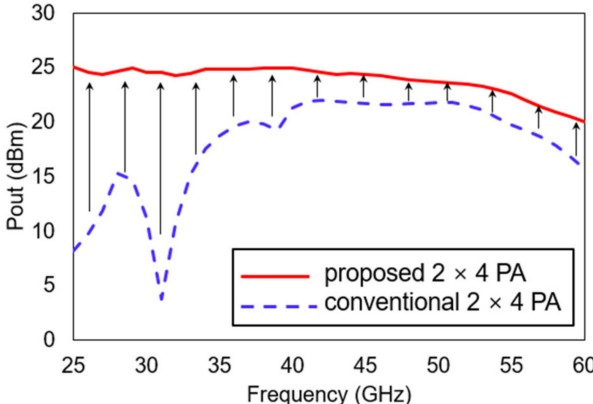

**Figure 6.** Output power comparison of the designed PAs according to frequency (input power = 15 dBm).

As the power divider/combiner causes extra insertion loss, the power combining efficiency is an important factor. We investigated it via a 2.5 dimensional electromagnetic (EM) simulation using Keysight's advanced design system (ADS) momentum tool. From 26 GHz to 56 GHz, the insertion loss of the input power divider was approximately 0.9 dB to 1.8 dB and that of the output power combiner was approximately 0.7 dB to 1.2 dB. Therefore, the calculated power combining efficiency was approximately 50% to 69% from 26 GHz to 56 GHz.

## 3. Implementation and Measured Results

### 3.1. Implementation

The proposed PA was designed as a 2 × 4 distributed power-combining structure and was fabricated using a commercial 0.15 μm GaAs pHEvMT MMIC process with $f_T$ of 85 GHz and $f_{MAX}$ of 190 GHz. Figures 7 and 8 show the chip photograph and schematic of the proposed PA, respectively.

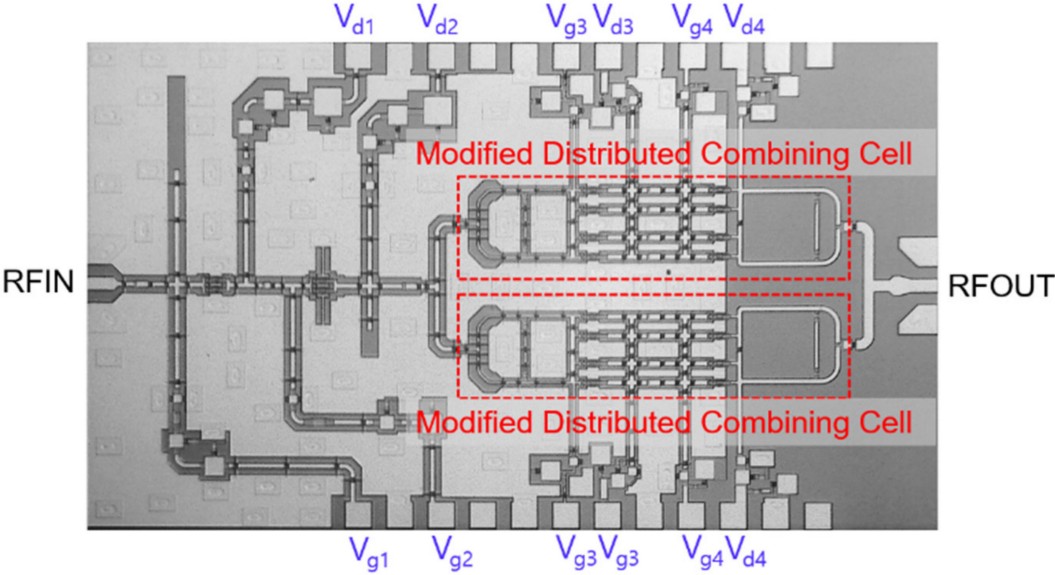

**Figure 7.** Die photograph of the proposed PA (chip size: 3.1 mm × 1.7 mm).

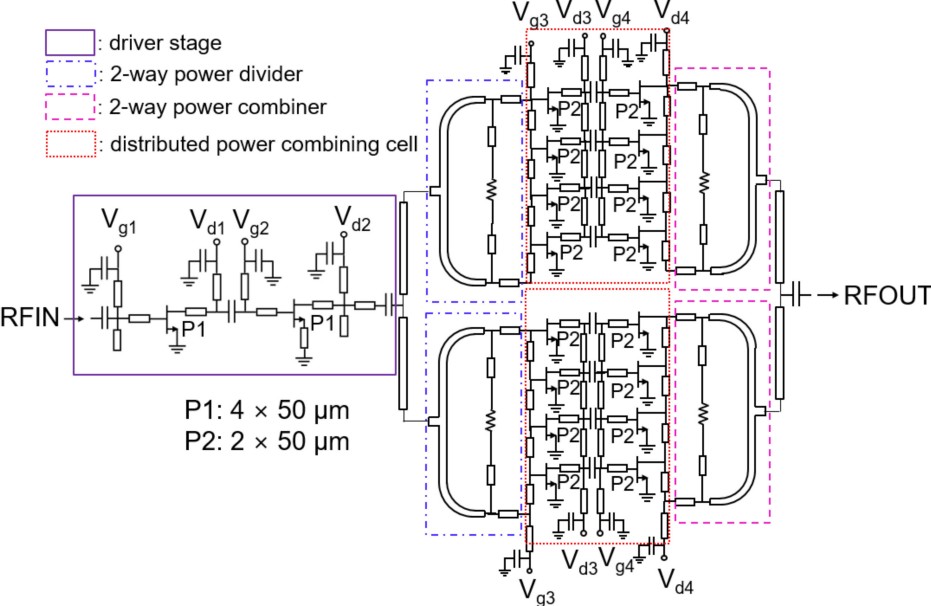

**Figure 8.** Schematic of the proposed PA.

The proposed PA consists of four stages. The preceding two stages were designed as common-source types in order to drive input power, and they used $4 \times 50$ μm FETs. The two-way modified distributed power combining structures ($N = 4$) followed as the main power stages. In order to boost the gain of the combining structure, they were designed as using $2 \times 50$ μm FETs. The proposed PA was almost designed as a grounded coplanar waveguide (GCPW) type, but the final output combining network was designed as the microstrip type to achieve low Q matching through wideband frequencies.

*3.2. Measured Results*

The measured S-parameter is shown in Figure 9, and the total current was 575 mA. The measured small signal gain was 13 to 25 dB from 25 GHz to 60 GHz. At 40 to 50 GHz, this gain exhibited a small dip which seemed to result from narrow band input matching by simple series lines and open stubs. If the distributed structure or feedback topology is applied to the input matching network, the gain flatness can be improved.

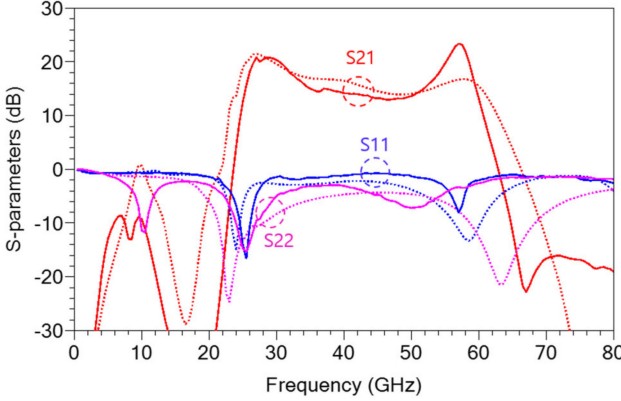

**Figure 9.** S-parameter of the proposed PA at $V_{d1} = 3$ V; $V_{d2} = V_{d4} = 4$ V; $V_{d3} = 2.5$ V; and $V_{g1\sim4} = -0.4$ V (solid line: measurement, dashed line: simulation).

Figure 10 represents the measured $P_{sat}$, input 1 dB compression point ($P_{1dB}$) and PAE according to frequency. The bias conditions during measurement were the same as the ones shown in Figure 9.

From 25 to 60 GHz, the measured $P_{sat}$ and $P_{1dB}$ were 15.4 to 24.2 dBm and 14.7 to 22.8 dBm, respectively. The measured PAEs at $P_{sat}$ and $P_{1dB}$ were 1.3% to 12.7% and 1.3% to 9.2%, respectively. The proposed PA shows the wideband output power characteristics from 26 to 56 GHz with 20.3 to 24.2 dBm.

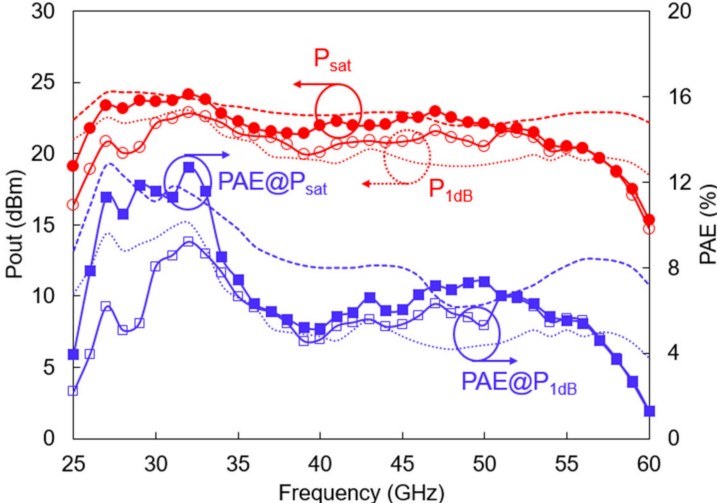

**Figure 10.** $P_{sat}$, $P_{1dB}$, and PAEs at $P_{sat}$ and $P_{1dB}$ of the proposed PA according to frequencies (solid line: measurement, dashed line: simulation).

Figure 11a,b show the measured gain, $P_{out}$, and PAE according to input power at 32 GHz and 47 GHz, which were the frequencies at which peak $P_{out}$ and peak PAE were obtained among the Ka and Q bands, respectively, in the proposed PA. At 32 GHz, the proposed PA obtained 24.2 dBm $P_{sat}$ and 12.7% peak PAE and at 47 GHz, it did 23 dBm $P_{sat}$ and 7.5% peak PAE, respectively.

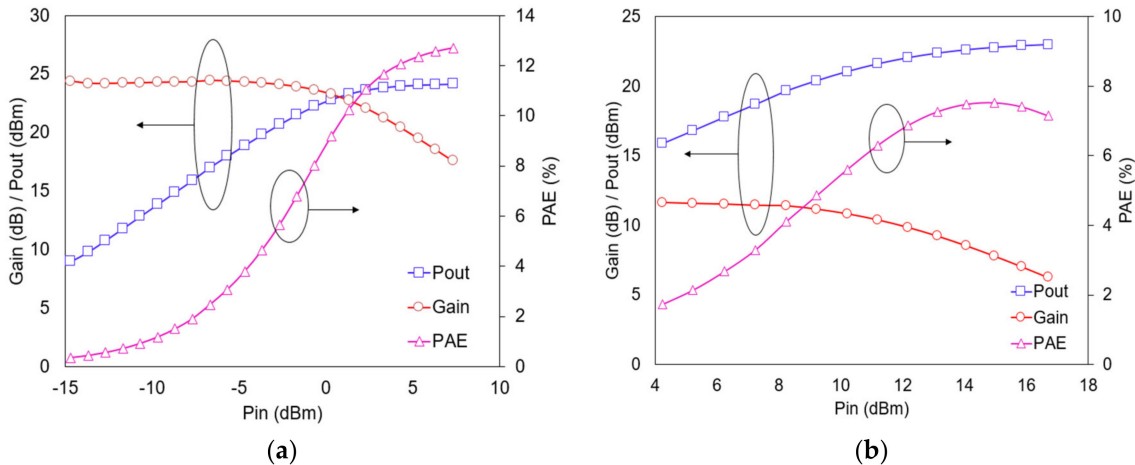

**Figure 11.** Gain, $P_{out}$, and PAE of the proposed PA according to input power at (**a**) 32 GHz and (**b**) 47 GHz.

Table 2 summarizes the performance of the reported broadband PAs available in the entire Ka/Q-band as well as the previous SEDFDC PAs. When compared with other previously reported broadband PAs, the proposed PA exhibited excellent power characteristics that could cover the full Ka-band and Q-band to the V-band with a $P_{sat}$ over 20 dBm. From the point of "power bandwidth product", this work showed the highest score among other previous works as shown in Table 2. In particular, it is noteworthy that the proposed PA obtained an approximately eight times higher power bandwidth product than Reference [4] which adopted a similar SEDFDC topology. This result

indicates that our modified SEDFDC can generate a high broadband power output in the in the millimeter wave band.

**Table 2.** Performance comparison of broadband millimeter-wave PA MMICs available in the entire Ka-band and Q-band/SEDFDC PA MMICs.

| Reference | Frequency (GHz) | Technology | $P_{sat}$ (dBm) | $P_{1dB}$ (dBm) | Peak PAE (%) | Power Bandwidth Product [1] | Gain (dB) | Topology |
|---|---|---|---|---|---|---|---|---|
| [10] | 29–57 | 28 nm CMOS | 15.1–16.6 | 10.9–13.4 | 24.2 | 0.91 | 20 | Transformer-based combining |
| [11] | 19–57 | 0.15 μm GaAs pHEMT | 16–19 | NA | NA | 1.51 | 15–27 | DA + cascade amp |
| [12] | 15–50 | 0.15 μm GaAs pHEMT | 18–22 | 15–19 | 19 | 2.21 | 15 | Cascaded DA |
| [13] | 20–60 | 45 nm SOI PMOS | 19–21.7 | 17.5–19 | 18.4 | 3.18 | 16 | Stacked DA |
| [4] | 53–62.8 | 65 nm CMOS | 16–17.2 | 15.5–16.6 | 11.3 | 0.39 | 14.5–17.5 | SEDFDC |
| [5] | 44–60 | 65 nm CMOS | 11 [2] | 9.7 [2] | 7.1 [2] | 0.20 | 8.3 [2] | SEDFDC |
| This work | 26–56 | 0.15 μm GaAs pHEMT | 20.3–24.2 | 19.9–22.8 | 12.7 | 3.21 | 12.9–21.8 | Modified SEDFDC |

[1] Power bandwidth product = $P_{sat\_min} \times (Freq._{max} - Freq._{min})$ (W×GHz). [2] Measured result at 52 GHz.

## 4. Conclusions

This work demonstrated millimeter-wave PAs that produced 20.3 dBm to 24.2 dBm $P_{out}$, 12.9 dB to 21.8 dB power gain, and 5.2% to 12.7% PAE within 26 GHz to 56 GHz. The proposed PA modified the SEDFDC design and used a GaAs pHEMT MMIC process to achieve an approximately eight times higher power bandwidth product than the previous Si CMOS-based SEDFDCs. The proposed method can increase the number of SEDFDCs while maintaining wideband power characteristics. Since this work can be applied to other technologies, such as gallium nitride (GaN) or CMOS in the form of massive combining, it is an attractive candidate for future broadband millimeter-wave power combining.

**Funding:** This research received no external funding.

**Acknowledgments:** This work was supported by the research grant of Pai Chai University in 2020.

**Conflicts of Interest:** The author declares no conflict of interest.

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
