# Peer review of "Broadband Millimeter-Wave Power Amplifier Using Modified 2D Distributed Power Combining"

_electronics, doi:10.3390/electronics9060899_

Round 1
Reviewer 1 Report
Very well organized paper.
1) Figure 4 (a,c,d) , it is preferred to have the vertical axis numbering finished at lower values. Ex: Fig. 4a, end at 10 instead of 25, Fig. 4c, end at 6 instead of 18 and so on... This will help in better clarification of results.
2) More discussion of results is needed after Figs. 3 and 4 showing the importance of obtained results.
3) In the table of comparison (Table 2), the frequency range is NOT unified. So, how can your work be compared with others?
Also, in the same table, I prefer to add a column showing the percentage of improvement achieved by your work compared to others.
4) The conclusion section must contain the main (numerical) results and improvement achieved. Some of them is found in the abstract.
5) References 8+9 need the place where the conference (symposium) held.
Author Response
I would like to thank the reviewers for your valuable comments. I did my best to fully reflect the reviewers' comments in the revised manuscript. Thank you again for the opportunity to make our paper more complete.

Reviewer 2 Report
- The paper presents a 2D distributed power combining structure using 0.15 um GaAs phemt technology. The structure is based on CRLH transmission line structures integrated with dual-fed distributed amplifiers to achieve the power combining. Some related works should be referenced and cited, for instance:
- Wu, Chung-Tse Michael, and Tatsuo Itoh. "Dual-fed distributed amplifier-based CRLH-leaky wave antenna for gain-enhanced power combining." In 2012 IEEE MTT-S International Microwave Workshop Series on Innovative Wireless Power Transmission: Technologies, Systems, and Applications, pp. 87-90. IEEE, 2012.
- Wu, Chung-Tse Michael, Yuandan Dong, Jim S. Sun, and Tatsuo Itoh. "Ring-resonator-inspired power recycling scheme for gain-enhanced distributed amplifier-based CRLH-transmission line leaky wave antennas." IEEE transactions on microwave theory and techniques 60, no. 4 (2012): 1027-1037.
- The 2D CRLH based power combining scheme has also been proposed in the following article: Fei, Wei, Hao Yu, Yang Shang, and Kiat Seng Yeo. "A 2-D distributed power combining by metamaterial-based zero phase shifter for 60-GHz power amplifier in 65-nm CMOS." IEEE Transactions on Microwave Theory and Techniques 61, no. 1 (2012): 505-516. The author should compare their performances such as gain, PAE, power consumption, bandwidth, chip size, etc.
- As the power divider/combiner will introduce extra insertion loss, what is the power combining efficiency?
Author Response

(The authors gave the same response as above.)
